# The effect of gender and parenting daughters on judgments of morally controversial companies

Paweł Niszczota[1]*, Michał Białek[2]

1 Department of International Finance, Poznań University of Economics and Business, Poznań, Poland,
2 Institute of Psychology, University of Wrocław, Wrocław, Poland

* pawel.niszczota@ue.poznan.pl

## Abstract

Earlier findings suggest that men with daughters make judgments and decisions somewhat in line with those made by women. In this paper, we attempt to extend those findings, by testing how gender and parenting daughters affect judgments of the appropriateness of investing in and working for morally controversial companies ("sin stocks"). To do so, in Study 1 ($N = 634$) we investigate whether women judge the prospect of investing in sin stocks more harshly than men do, and test the hypothesis that men with daughters judge such investments less favorably than other men. In Study 2 ($N = 782$), we investigate the willingness to work in morally controversial companies at a significant wage premium. Results show that—for men—parenting daughters yields harsher evaluations of sin stocks, but no evidence that it lowers the propensity to work in such companies. This contrasts to the effect of gender: women reliably judge both investment and employment in morally controversial companies more harshly than men do. We suggest that an aversion towards morally controversial companies might be a partial determinant of the gender gap in wages.

## 1. Introduction

Purely self-interested economic decisions would be driven solely by expected gains (conditional on risk), but in reality people also consider externalities—or the interest of others—when making their decisions (e.g., [1–3]). The extent to which people consider these externalities varies across individuals, with some weighting them more than others. A recent study [4] has shown that women (relative to men) judge investment in morally controversial companies —better known in the financial literature as "sin" stocks [5, 6]–to be less morally appropriate. Assuming there is a premium from investing in sin stocks [6] and women are more likely to exclude them from portfolios, this would lead women to have lower returns from their stock portfolios (e.g., in pension funds, mutual funds, or stocks held directly) than men. Another example of increased sensitivity to externalities or other-preferences is the existence of a "daughter effect" in corporate decisions: firms in which the CEO has a daughter have higher corporate social responsibility scores [7], and hire more women to their boards [8, 9],

**Data Availability Statement:** All studies have been pre-registered. The pre-registration documents, data and materials are available at: https://osf.io/juk86.

**Funding:** This research has been supported by grant 2018/31/D/HS4/01814 awarded to PN, from

the National Science Centre, Poland (Narodowe
Centrum Nauki; https://www.ncn.gov.pl). The
funders had no role in study design, data collection
and analysis, decision to publish, or preparation of
the manuscript.

**Competing interests:** The authors have declared
that no competing interests exist.

compared to firms in which the CEO does not have a daughter. Crucially, such effects are not
related to parenting sons, suggesting that this effect is driven by the internalization of prefer-
ences of children of a different gender than that of the decision maker, and is not due to having
children in general.

In this paper we decided to further investigate the gender and daughter effects in economic
decision making. Our study can be seen as in accordance with the so-called creative destruc-
tion approach to replication, in which experiments are replicated but boundary conditions
and alternative explanations are also tested [10]. Firstly, we extend the findings concerning
gender differences in investment decisions [4] to the willingness to work in a morally contro-
versial company for a higher wage. While investment in equities is consequential, the effects
are limited to people who invest a significant portion of their income in stocks. A more conse-
quential decision concerns the prospect of working for a morally controversial company [11].
Not only are employment considerations relevant to a larger number of individuals, they affect
wealth to a much greater extent if working for such a company will be the main source of their
income. The importance of moral concerns in economic decisions can be a function of the size
of incentives [12] and their likelihood [13]. Hence, a more consequential choice can be a more
concrete test for the gender differences in morally-driven economic decision making.

Secondly, we contribute to the strand of literature investigating "daughter effects", in which
some issues remain unresolved. While Cronqvist and Yu [7] suggest that the daughter effect is
due to female socialization [14] of the (mostly male) CEOs, they do not test whether the effect
of daughters on the policies of male and female CEOs is different (an alternative is the exis-
tence of a "son effect" for women, due to analogous male socialization). Moreover, extant
research reports the daughter effect in upper echelons of corporate executives [7] and politi-
cians [14], but not in laypeople. Obviously, CEOs and politicians are not a representative sam-
ple of the general population, because people with particular characteristics and personality
traits strive for such prestigious positions [15]. Also, parents might make different "invest-
ment" in daughters and sons if their social status is higher [16, 17].

The main explanation for why parenting a daughter can affect behavior concerns a shift in
the attitudes that people go through while having a daughter. One piece of evidence suggests
that parents' preferences change as soon as parents find out about the gender of their child: a
study performed in hospitals suggests that future parents become more risk-averse when they
find out that their child will be female [18]. Most of this effect happens after parents find out
about the sex of their unborn child; the shift in risk aversion largely carries over to after the
child is born. Considering that it is present across both men and women, it is hypothesized by
Pogrebna et al. [18] to have a non-biological basis (e.g., not be due to hormonal shifts during
pregnancy). Here, we assume that a similar effect will be at play, meaning that if the effect of
having a child on parents' preferences depends on the child's sex, the effects will be immediate,
rather than gradual. In other words, parents will be able to internalize the expected future
experiences of their children. In pioneering studies on the effect of the gender of the child,
Warner [19] argued that this is out of self-interest: men and women cognizant of how their
child will be treated in society are more likely to adopt a more feminist (egalitarian) worldview
if their child is a daughter. Such worldview shifts can plausibly spillover to other (e.g., social or
risk) preferences [7].

Below, we report two studies with a very similar design. In Study 1, participants rated their
willingness to invest in morally controversial companies, and in Study 2 they rated their will-
ingness to work in such companies. We tried to predict these decisions by the gender of the
decision makers and the gender of their children. For brevity, we report their findings jointly,
including cross-study comparisons. In both cases, the designs correspond to extensive mar-
gins, i.e., we are asking participants to decide how acceptable the *act* of investment or

employment is to them, and not intensive margins, i.e., those that would relate to an *increase* of investment or labor (e.g., extending the number of working hours). In other words, we ask about whether people would be willing to invest and be employed in selected companies at all. Thus, we assumed that participants were able to distinguish between extensive and intensive margins, even if they coincidentally did already invest or work in any of the companies from the listed industries.

## 2. Data and methodology

### 2.1 Data

We recruited Mechanical Turk participants that were residents of the United States, had at least 95% tasks approved, and completed at least 1,000 of them. Prior to data collection, we performed power analyses. We collected data to ensure 95% power to detect an $f^2 = 0.02$ effect on the subset of participants with biological children (pilot studies indicated that roughly 90% of Mechanical Turk participants with children had solely biological children). Considering an expected exclusion rate of 10–20% based on the attention check [20], we collected data from 805 (Study 1) and 906 participants (Study 2). Ultimately, using a preregistered exclusion criterion, we analyzed $N = 634$ participants in Study 1 (having excluded 168 participants), and $N = 782$ in Study 2 (having excluded 124 participants). Altogether, we analyzed data from 1,416 participants: 650 women and 766 men. S1 Table describes the sample using detailed demographic information.

### 2.2 Methodology

In Study 1, we have asked participants to consider investment in seven morally controversial industries (involved in abortion/abortifacients, adult entertainment, animal testing, controversial weapons, fur, gambling, tobacco) and seven conventional industries (involved in air freight/logistics, construction/engineering, household durables, marine transport, road/rail transport, semiconductors/semiconductor equipment, water utilities); these were used in previous research [4, 21], but to aid readability the descriptions are provided in the S1 Appendix. More specifically, participants assessed the moral appropriateness of investing in industries (*Would it be morally appropriate if you invested in companies from the following industries*?), on a scale of 1 (*not at all*) to 7 (*completely*). The dependent variable was the difference between the mean sin stock judgment and the mean conventional stock judgment. In Study 2, participants assessed their willingness to switch employment from a non-controversial industry to employment in morally controversial industries. They were told to imagine that such a decision would lead to a 25% increase in their wage. The dependent variable was the mean of judgments of five industries: in accordance with the preregistration, we excluded abortion/abortifacient and adult entertainment industries from the employment sample, because of potential gender differences in their evaluation as employers.

Note the differences between the two studies: investment, but not employment, is easily reversible; for regular individuals the economic gain from the immoral choice is larger in the employment scenario; finally, the increased benefit when opting for the immoral industry is only explicitly mentioned in the employment scenario.

We created a series of models. To test for a gender effect, and a daughter (or son) effect across participants' gender, we estimated the following model:

$$Y_i = \alpha + \beta_1 gender_i + \beta_2 [daughters > 0]_i + \beta_3 [sons > 0]_i + \beta_4 controls_i + \varepsilon_i \qquad (1)$$

Next, we conducted regressions containing interactions, with the intention of investigating whether the daughter effect is different across genders:

$$Y_i = \alpha + \beta_1 gender_i + \beta_2 [daughters > 0]_i + \beta_3 [sons > 0]_i \\ + \beta_4 gender_i \times [daughters > 0]_i + \beta_5 gender_i \times [sons > 0]_i + \beta_6 controls_i + \varepsilon_i \quad (2)$$

Finally, consistent with Cronqvist and Yu, we attempted to decompose the effect of the first and subsequent daughters and sons, in specifications that excluded and included interaction terms:

$$Y_i = \alpha + \beta_1 gender_i + \beta_2 [daughters > 0]_i + \beta_3 [sons > 0]_i \\ + \beta_4 [number\ of\ subsequent\ daughters]_i \\ + \beta_5 [number\ of\ subsequent\ sons]_i + \beta_6 controls_i + \varepsilon_i \quad (3)$$

$$Y_i = \alpha + \beta_1 gender_i + \beta_2 [daughters > 0]_i + \beta_3 [sons > 0]_i \\ + \beta_4 [number\ of\ subsequent\ daughters]_i + \beta_5 [number\ of\ subsequent\ sons]_i + \beta_6 gender_i \\ \times [daughters > 0]_i + \beta_7 gender_i \times [sons > 0]_i + \beta_8 gender_i \\ \times [number\ of\ subsequent\ daughters]_i + \beta_9 gender_i \times [number\ of\ subsequent\ sons]_i \\ + \beta_{10} controls_i + \varepsilon_i \quad (4)$$

We controlled for the effect of age, education level, household size, household income size, marital status, self-rated general risk tolerance of the participants [22], self-rated investment knowledge, and actual basic investment knowledge, assessed with six items used in Niszczota and Białek [4].

The studies were preregistered (https://aspredicted.org/jb4fk.pdf; https://aspredicted.org/md29b.pdf) and approved by the Committee of Ethical Research conducted with participation of humans at the Poznań University of Economics and Business (Resolution 7/2020). Informed consent was obtained from all participants. Data and materials are deposited at a publicly available repository (https://osf.io/juk86).

# 3. Results

## 3.1 Descriptive statistics

Basic demographic data and data concerning participants' children is presented in S1 Table. Men and women in our samples reported similar age, household income levels, and employment status. As in prior studies, women were less risk tolerant (e.g., [22]) and rated themselves as less knowledgeable in investment matters, although actual differences—measured using six items—were small [4]. Fig 1 shows the distribution of judgments of stocks in Study 1, separately for men and women, and separately for those who do and who don't have daughters. As expected, sin stocks were judged to be significantly less morally appropriate investment propositions than conventional stocks. Men with daughters judged sin stocks to be less appropriate investment propositions, and at the same time judged conventional stocks to be more morally appropriate propositions. This led to an increase in the discrepancy between judgments of investment in sin versus conventional stocks. Interestingly, the effect was opposite in women: women with daughters judged sin stocks more favorably and conventional stocks less favorably than women without daughters, which lead to a decrease in the sin stock bias.

## 3.2 The gender and daughter effect

In the majority of analyses—as preregistered—we separately tested data from entire samples (*All*), and samples where non-biological children were excluded (*Biological*). While Cronqvist

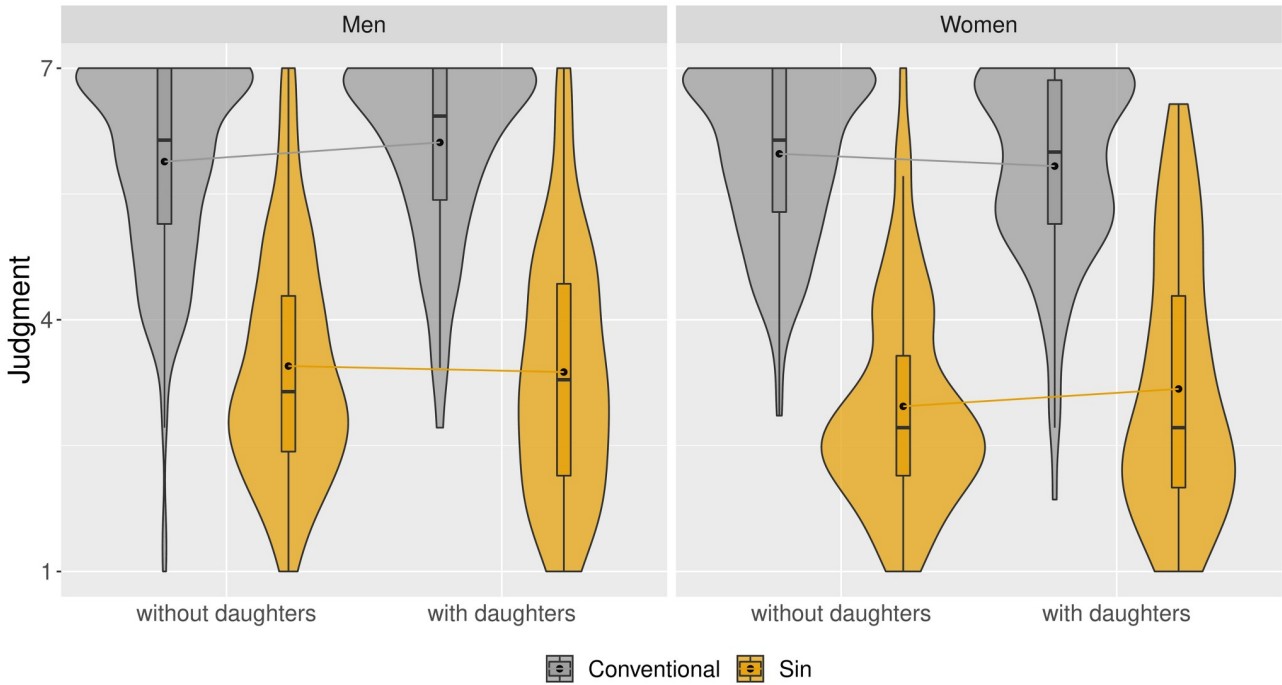

**Fig 1. Judgment of investments in morally controversial (sin) and conventional industries (Study 1).** *Notes*: Dots represent means.

and Yu [7] argue that having a daughter should affect the judgments and decisions of men regardless of whether their child is biological or not (i.e., adopted, a stepchild or a foster child), it is worthwhile to reinvestigate such claims. Moreover, it can be argued that the gender of non-biological children is not (entirely) exogenous, whereas in the case of biological children it is reasonable to assume that it is.

Tests of potential gender and daughter effects in the judgments of companies are presented in Table 1. In Panel A, we pooled the data from men and women. Consistent with [4], women judge investment in morally controversial companies to be less appropriate than men do. Our results also show that women judge the perspective of employment in morally controversial companies less favorably. Although there is no daughter effect in the entire sample, this would be expected if there were large differences between having daughters between men and women, which—as is evident from Fig 1—is indeed the case.

In Panel B, we present results of regressions containing interactions. This allows us to test two hypotheses. Firstly, this allows us to test whether the daughter effect is significant in men (which is shown via the coefficient for *Daughters > 0* (the single term), given that we coded men as zero). Secondly, this allows us to test whether this effect is different in women (which is shown via the *Gender × [Daughters > 0]* interaction term). Our results replicated the gender bias in moral economic decision making, and show that men parenting daughters judge investment in morally controversial companies (but not employment in them) as less appropriate than men without daughters. In contrast, there is no such effect for sons, suggesting that the daughter effect is not the result of having children, but having daughters. Interaction terms suggest that the effect of having daughters is significantly different in women, consistent with what is shown in Fig 1. There are no differences in the effects of having sons on judgments, suggesting that, overall, having sons does not affect the judgments of neither men nor women. Additionally, tests of the size of coefficients show that for men that were given the hypothetical

**Table 1.** The daughter effect and differences in this effect between men and women.

| | Investment | | Employment | |
|---|---|---|---|---|
| | *All* | *Biological* | *All* | *Biological* |
| *Panel A. Entire sample analysis* | | | | |
| Gender (0 = *m*, 1 = *f*) | −0.31 ** | −0.35 *** | −0.27 ** | −0.27 ** |
| | (0.12) | (0.13) | (0.11) | (0.12) |
| Daughters > 0 | −0.12 | −0.07 | −0.16 | −0.14 |
| | (0.15) | (0.15) | (0.14) | (0.15) |
| Sons > 0 | 0.17 | 0.11 | −0.01 | −0.01 |
| | (0.16) | (0.16) | (0.13) | (0.14) |
| Risk tolerance | 0.17 *** | 0.17 *** | 0.12 *** | 0.13 *** |
| | (0.03) | (0.03) | (0.02) | (0.03) |
| Objective investment knowledge | -0.29 *** | -0.28 *** | -0.11 *** | -0.09 ** |
| | (0.04) | (0.05) | (0.04) | (0.04) |
| Subjective investment knowledge | 0.09 * | 0.08 * | 0.17 *** | 0.16 *** |
| | (0.05) | (0.05) | (0.04) | (0.04) |
| Marital status: married | 0.23 | 0.23 | 0.40 *** | 0.39 ** |
| | (0.18) | (0.18) | (0.15) | (0.15) |
| Marital status: divorced or widowed | 0.19 | 0.19 | -0.20 | -0.17 |
| | (0.25) | (0.26) | (0.20) | (0.20) |
| Education: doctoral level or equivalent | 0.52 | 0.68 | -0.70 | -0.91 ** |
| | (0.50) | (0.56) | (0.47) | (0.44) |
| Education: Master's degree or equivalent | 0.45 ** | 0.37 ** | -0.12 | -0.10 |
| | (0.18) | (0.18) | (0.17) | (0.17) |
| Education: primary school | -0.35 | -0.34 | 0.22 | 0.25 |
| | (0.38) | (0.38) | (0.26) | (0.27) |
| Education: secondary school | -0.31 ** | -0.31 ** | 0.10 | 0.08 |
| | (0.15) | (0.16) | (0.12) | (0.13) |
| Employment: self-employed | 0.13 | 0.14 | -0.65 *** | -0.62 *** |
| | (0.17) | (0.18) | (0.15) | (0.16) |
| Employment: unemployed | -0.25 | -0.26 | -0.45 *** | -0.49 *** |
| | (0.20) | (0.20) | (0.14) | (0.14) |
| Age (logged) | -1.12 *** | -1.05 *** | -0.02 | -0.07 |
| | (0.27) | (0.28) | (0.20) | (0.20) |
| Household income (midpoint, logged) | -0.17 * | -0.14 | -0.34 *** | -0.33 *** |
| | (0.10) | (0.10) | (0.09) | (0.09) |
| *N* | 634 | 604 | 781 | 742 |
| Adjusted R$^2$ | 0.248 | 0.229 | 0.186 | 0.188 |
| *Daughters vs sons* | *p* = 0.18 | *p* = 0.44 | *p* = 0.45 | *p* = 0.51 |
| *Panel B. Analysis of gender differences in daughter effect* | | | | |
| Gender (0 = *m*, 1 = *f*) | −0.49 *** | −0.46 *** | −0.27 * | −0.25 * |
| | (0.16) | (0.16) | (0.14) | (0.14) |
| Daughters > 0 | −0.48 ** | −0.41 * | −0.12 | −0.05 |
| | (0.22) | (0.24) | (0.20) | (0.21) |
| Sons > 0 | 0.20 | 0.19 | −0.05 | −0.05 |
| | (0.21) | (0.22) | (0.19) | (0.20) |
| Gender × [Daughters > 0] | 0.66 ** | 0.62 ** | −0.06 | −0.15 |
| | (0.28) | (0.30) | (0.24) | (0.25) |

*(Continued)*

**Table 1.** (Continued)

| | Investment | | Employment | |
|---|---|---|---|---|
| | *All* | *Biological* | *All* | *Biological* |
| Gender × [Sons > 0] | −0.08 | −0.19 | 0.06 | 0.07 |
| | (0.26) | (0.27) | (0.24) | (0.25) |
| Risk tolerance | 0.17 *** | 0.17 *** | 0.12 *** | 0.13 *** |
| | (0.03) | (0.03) | (0.02) | (0.03) |
| Objective investment knowledge | -0.29 *** | -0.28 *** | -0.11 *** | -0.09 ** |
| | (0.04) | (0.05) | (0.04) | (0.04) |
| Subjective investment knowledge | 0.09 * | 0.08 * | 0.17 *** | 0.16 *** |
| | (0.05) | (0.05) | (0.04) | (0.04) |
| Marital status: married | 0.25 | 0.25 | 0.40 *** | 0.40 ** |
| | (0.18) | (0.18) | (0.15) | (0.16) |
| Marital status: divorced or widowed | 0.21 | 0.18 | -0.20 | -0.18 |
| | (0.25) | (0.26) | (0.20) | (0.20) |
| Education: doctoral level or equivalent | 0.51 | 0.64 | -0.69 | -0.91 ** |
| | (0.52) | (0.56) | (0.47) | (0.44) |
| Education: Master's degree or equivalent | 0.47 *** | 0.39 ** | -0.12 | -0.10 |
| | (0.18) | (0.18) | (0.17) | (0.17) |
| Education: primary school | -0.33 | -0.32 | 0.22 | 0.24 |
| | (0.38) | (0.38) | (0.26) | (0.27) |
| Education: secondary school | -0.31 ** | -0.30 * | 0.10 | 0.08 |
| | (0.15) | (0.16) | (0.12) | (0.13) |
| Employment: self-employed | 0.12 | 0.12 | -0.64 *** | -0.62 *** |
| | (0.17) | (0.18) | (0.15) | (0.16) |
| Employment: unemployed | -0.25 | -0.26 | -0.45 *** | -0.50 *** |
| | (0.20) | (0.20) | (0.14) | (0.14) |
| Age (logged) | -1.10 *** | -1.02 *** | -0.02 | -0.06 |
| | (0.27) | (0.28) | (0.20) | (0.21) |
| Household income (midpoint, logged) | -0.17 * | -0.13 | -0.34 *** | -0.33 *** |
| | (0.10) | (0.10) | (0.09) | (0.09) |
| *N* | 634 | 604 | 781 | 742 |
| Adjusted R² | 0.252 | 0.232 | 0.184 | 0.186 |
| *Daughters vs sons (for men)* | *p = 0.03* | *p = 0.07* | *p = 0.78* | *p = 0.99* |

*Notes*: Robust standard errors are in parentheses. Reference points for control variables: single (marital status), Bachelor's degree (education), employee (employment).

\*\*\* $p < 0.01$

\*\* $p < 0.05$

\* $p < 0.1$.

investment scenario, the effect of having daughters is indeed different from the effect of having sons.

Analyses of the prospective employment in morally controversial companies once again point to the existence to a gender effect in judgment. There was, however, no evidence of a daughter effect in men, and no evidence that the effect of having daughters was different in women. Similarly, having sons affected judgments of neither male nor female participants.

To dissect the effect of gender and having children on the judgment of morally controversial companies, we prepared our data for mixed-model analyses. Table 2 shows estimates of a linear mixed-model using a pooled dataset from Studies 1 and 2 (and for both studies

**Table 2. Mixed model: Five overlapping morally controversial industries, used both in Study 1 and Study 2.**

| | Judgment | | |
|---|---|---|---|
| | *Studies 1 and 2* | *Study 1* | *Study 2* |
| *Panel A. Entire sample analysis* | | | |
| Gender (0 = *m*, 1 = *f*) | −0.26 *** | −0.30 *** | −0.27 ** |
| | (0.08) | (0.11) | (0.11) |
| Daughters > 0 | −0.10 | −0.06 | −0.16 |
| | (0.09) | (0.13) | (0.13) |
| Sons > 0 | 0.10 | 0.21 | −0.01 |
| | (0.09) | (0.14) | (0.13) |
| Judgment type (0 = *employment*, 1 = *investment*) | −0.11 | | |
| | (0.17) | | |
| Risk tolerance | 0.14 *** | 0.17 *** | 0.12 *** |
| | (0.02) | (0.03) | (0.02) |
| Objective investment knowledge | -0.14 *** | -0.18 *** | -0.11 *** |
| | (0.03) | (0.04) | (0.04) |
| Subjective investment knowledge | 0.19 *** | 0.20 *** | 0.17 *** |
| | (0.03) | (0.04) | (0.03) |
| Marital status: married | 0.32 *** | 0.23 | 0.40 *** |
| | (0.10) | (0.15) | (0.14) |
| Marital status: divorced or widowed | -0.00 | 0.25 | -0.20 |
| | (0.16) | (0.23) | (0.22) |
| Education: doctoral level or equivalent | 0.22 | 1.09 *** | -0.70 * |
| | (0.29) | (0.41) | (0.40) |
| Education: Master's degree or equivalent | -0.00 | 0.22 | -0.12 |
| | (0.12) | (0.19) | (0.15) |
| Education: primary school | 0.10 | -0.27 | 0.22 |
| | (0.20) | (0.32) | (0.26) |
| Education: secondary school | -0.02 | -0.13 | 0.10 |
| | (0.09) | (0.13) | (0.12) |
| Employment: self-employed | -0.31 *** | 0.03 | -0.65 *** |
| | (0.11) | (0.15) | (0.16) |
| Employment: unemployed | -0.35 *** | -0.19 | -0.45 *** |
| | (0.11) | (0.18) | (0.14) |
| Age (logged) | -0.10 | -0.24 | -0.02 |
| | (0.15) | (0.23) | (0.20) |
| Household income (midpoint, logged) | -0.24 *** | -0.11 | -0.34 *** |
| | (0.06) | (0.09) | (0.08) |
| *Random effects* | | | |
| $\sigma 2$ | 2.16 | 1.77 | 2.48 |
| $\tau 00$ | $1.60_{id}$ | $1.41_{id}$ | $1.51_{id}$ |
| | $0.48_{sin}$ | $0.17_{sin}$ | $0.48_{sin}$ |
| ICC | 0.46 | 0.47 | 0.45 |
| *N* | 7,075 | 3,170 | 3,905 |
| Marginal $R^2$ / Conditional $R^2$ | 0.121 / 0.525 | 0.178 / 0.565 | 0.100 / 0.501 |
| *Panel B. Analysis of gender differences in daughter effect* | | | |
| Gender (0 = *m*, 1 = *f*) | −0.32 *** | −0.42 *** | −0.27 * |
| | (0.10) | (0.15) | (0.14) |

*(Continued)*

**Table 2.** (Continued)

| | Judgment | | |
|---|---|---|---|
| | *Studies 1 and 2* | *Study 1* | *Study 2* |
| Daughters > 0 | −0.18 | −0.25 | −0.12 |
| | (0.13) | (0.19) | (0.18) |
| Sons > 0 | 0.08 | 0.18 | −0.05 |
| | (0.13) | (0.19) | (0.18) |
| Judgment type (0 = *employment*, 1 = *investment*) | −0.11 | | |
| | (0.17) | | |
| Gender × [Daughters > 0] | 0.16 | 0.35 | −0.06 |
| | (0.17) | (0.24) | (0.24) |
| Gender × [Sons > 0] | 0.03 | 0.04 | 0.06 |
| | (0.17) | (0.24) | (0.23) |
| Risk tolerance | 0.14 *** | 0.17 *** | 0.12 *** |
| | (0.02) | (0.03) | (0.02) |
| Objective investment knowledge | -0.14 *** | -0.18 *** | -0.11 *** |
| | (0.03) | (0.04) | (0.04) |
| Subjective investment knowledge | 0.19 *** | 0.19 *** | 0.17 *** |
| | (0.03) | (0.04) | (0.03) |
| Marital status: married | 0.32 *** | 0.26 * | 0.40 *** |
| | (0.10) | (0.15) | (0.14) |
| Marital status: divorced or widowed | 0.01 | 0.26 | -0.20 |
| | (0.16) | (0.24) | (0.22) |
| Education: doctoral level or equivalent | 0.23 | 1.09 *** | -0.69 * |
| | (0.29) | (0.41) | (0.40) |
| Education: Master's degree or equivalent | 0.00 | 0.24 | -0.12 |
| | (0.12) | (0.19) | (0.15) |
| Education: primary school | 0.11 | -0.26 | 0.22 |
| | (0.20) | (0.32) | (0.26) |
| Education: secondary school | -0.03 | -0.14 | 0.10 |
| | (0.09) | (0.13) | (0.13) |
| Employment: self-employed | -0.32 *** | 0.02 | -0.64 *** |
| | (0.11) | (0.15) | (0.16) |
| Employment: unemployed | -0.35 *** | -0.19 | -0.45 *** |
| | (0.11) | (0.18) | (0.14) |
| Age (logged) | -0.10 | -0.22 | -0.02 |
| | (0.15) | (0.23) | (0.20) |
| Household income (midpoint, logged) | -0.24 *** | -0.11 | -0.34 *** |
| | (0.06) | (0.09) | (0.08) |
| *Random effects* | | | |
| $\sigma 2$ | 2.16 | 1.77 | 2.48 |
| $\tau 00$ | $1.61_{id}$ | $1.41_{id}$ | $1.51_{id}$ |
| | $0.48_{sin}$ | $0.17_{sin}$ | $0.48_{sin}$ |
| ICC | 0.46 | 0.47 | 0.45 |
| *N* | 7,075 | 3,170 | 3,905 |

(*Continued*)

**Table 2.** (Continued)

| | Judgment | | |
|---|---|---|---|
| | *Studies 1 and 2* | *Study 1* | *Study 2* |
| Marginal R$^2$ / Conditional R$^2$ | 0.121 / 0.525 | 0.179 / 0.566 | 0.100 / 0.501 |

*Notes*: This table reports linear mixed-effects analyses. The fixed effects were the same as those reported in Eqs 1 and 2, with the addition of judgment type (investment or employment) for a pooled analysis of Studies 1 and 2. As random effects, we entered intercepts for subjects and items (industries), as well as by-subject and by-item random slopes for the effect of judgment type (Studies 1 and 2). Computations were performed using the *lmer* function from package *lmerTest* [23], which alters the function sourced from the *lme4* package in *R* [24]. Reference points for control variables: single (marital status), Bachelor's degree (education), employee (employment).

*** $p < 0.01$

** $p < 0.05$

* $p < 0.1$.

individually). The pooled model is estimated on a dataset that contains ratings of five morally controversial industries that overlap in both studies, which gives us 7,075 observations (5 ratings × 1,415 participants; in Study 2, one person was excluded from analyses due to missing data on regressors).

Consistent with earlier findings, there is a gender effect, i.e. women judge morally controversial companies less harshly than men do. However, there is no evidence of a daughter effect overall (Panel A), and additionally there is no evidence of such an effect in men or a gender difference in this effect (Panel B). Recall that in Table 1, results for Study 1 (i.e., the investment sample) suggested that men with daughters judged morally controversial companies—relative to conventional companies—less favorably than men without daughters. In Table 2 the effect of having daughters is no longer statistically significant. This suggests that in the earlier analysis morally controversial companies were only judged less favorably relative to conventional companies, somewhat weakening the validity of the earlier finding. A robust finding would suggest that morally controversial companies are judged less favorably *and* are judged less favorably relative to conventional stocks. However, only the latter is supported by our data.

Finally, in Fig 2 we illustrate how judgments of the five controversial industries differ across men and women: it corroborates with the results presented earlier, suggesting that women—compared to men—judge morally controversial companies as less appropriate. This finding holds for both the perspective of investing in morally controversial companies, and being employed in them.

### 3.3 Additional insights

In S2 Table, we test whether parenting more than one daughter has an effect on judgments and hypothetical decisions. Cronqvist and Yu [7] show that while the daughter effect is largely the result of having daughters in itself (i.e., parenting one daughter is sufficient to produce an effect on CEOs' corporate social responsibility policies), subsequent daughters cause an additional (albeit smaller) effect. However, in our case there is no evidence of an effect of subsequent daughters, in neither investment nor employment judgments.

Given that the daughter effect appears to be limited to males, we performed a subsample analysis using the key preregistered specifications, based on the gender of the participant. In addition, we tested an alternative specification, in which the key independent variable was the number of daughters, controlling for the number of children [14, 25]. The results of these

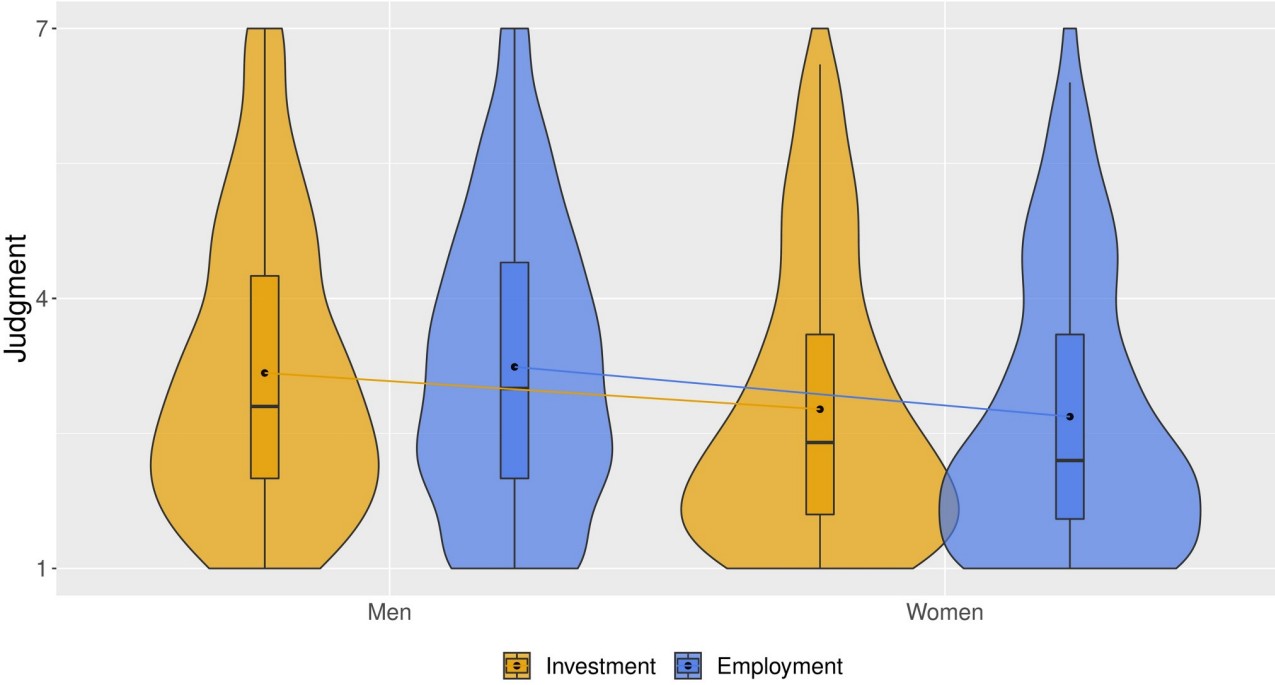

**Fig 2. Judgment of morally controversial industries that overlap in Studies 1 and 2.** *Notes*: Dots represent means.

analyses—presented in S3 Table—suggest that, for men, the daughter effect in investment is robust to an alternative specification.

S4–S6 Tables present the results of additional analyses. In S4 Table Panel A, we test the female socialization hypothesis, which suggests that the internalization of female preferences increases with the length of parenting. In other words, it is possible that the daughter effect is not instantaneous, i.e. it takes time for men to internalize the preferences (or adopt the perspective) of their daughters. To test this possibility, we logged the number of years that participants parented their children (after excluding the arguably less exogenous non-biological children). Our results do not support the hypothesis that the length of parenting affects judgments of controversial stocks. This suggests that it is more likely that the effect is merely the result of being a parent (see [19]). This is also consistent with the mechanism for the this and other daughter effects that was presented in the Introduction. Panel B investigates the effect of the gender of a child, in a subset of participants with only one biological child, which can be argued is a cleaner approach than investigating all children [7]. In this subsample, the daughter effect in men is of similar magnitude ($b_{\text{whole sample}} = -0.48$ vs $b_{\text{subsample}} = -0.41$), but it is not statistically significant from zero ($p = 0.24$). Note, however, that the subsample is 71.8% of the size of the entire sample, which increases measurement error and lowers statistical power.

In S5 Table we show how the daughter effect differs in participants will low and high household income levels, with the threshold arbitrarily set at $20,000 per person in a household. There is some indication that the effect of parenting a daughter has a different effect on employment for those with lower incomes ($b_{low\ income} = 0.10$ vs $b_{high\ income} = -0.16$). However, a three-way interaction (untabulated) does not suggest that these coefficients are significantly different from each other.

In S6 Table we show how participants judged each of the five overlapping morally controversial industries in Studies 1 and 2. While ratings of these industries show good internal

consistency (Cronbach's $\alpha$ = 0.82), it is possible that men with daughters will more harshly judge just a subset of these. Consistent with a gender effect, in all cases bar one (gambling industry) women judged morally controversial industries less favorably then men did. In contrast, only in one industry—animal testing—results were consistent with the existence of a daughter effect in men, albeit judgments of the fur industry made by women and of men with daughters also appeared to somewhat align. But note also that the issue relating to lower statistical power is exacerbated in such subsample analyses.

## 4. Discussion

The purpose of this research was to extend the findings concerning gender differences in moral or economic preferences (e.g., [4, 26]), and to provide a conceptual replication and extension of studies on the "daughter effect". This effect suggests that men might shift their behavior when parenting (or even finding out that they will parent in the future) a daughter, making their judgments and behaviors more in line with those manifested by women (e.g., [7, 14]).

Our findings show that women (relative to men) judge investment in morally controversial companies to be less morally appropriate, which replicates earlier results [4]. In addition, women state that they would be less willing to switch their place of employment to a morally controversial company, even though it would carry a 25% wage premium. This suggests that women consistently show an aversion towards controversial companies, as it was present in judgments and hypothetical decisions relating to financial and labor-market choices. This finding is consistent with the notion that people more concerned with morality (e.g., women [26]) are more reluctant to invest in controversial companies [27].

There are several insights from our work. Firstly, we investigate laypeople instead of people of high social status, such as CEOs [7–9], members of congress [14], or judges [28]. This would be consequential if parental investment in sons and daughters might depend on the social status of the parent [17, 18]. Studying laypeople makes our findings more relevant to the general population, and to more common decisions (e.g., concerning what mutual funds to invest in). Secondly, our models are aimed at directly testing whether the effect of parenting daughters is different across men and women. This would be expected from the female socialization hypothesis [14]: parenting daughters might make the preferences of men more similar to those exhibited by women, as it would help them adopt alternative perspectives on issues in which the opinions of men and women might differ. Yet, they would not cause a shift in the preferences of women, as they have the same gender as their daughters. Our findings show that parenting daughters leads to harsher evaluations of morally controversial investments, but only in men. In fact, women parenting a daughter judge morally controversial investments more favorably than women without daughters, a somewhat unexpected finding.

Our results showed a boundary condition of the daughter effect. In our case, a full conceptual replication of the findings of Cronqvist and Yu [7] would translate into a more negative view of morally controversial companies as investment propositions, and a lower willingness to be employed in such companies (at a significant premium). We observed the daughter effect in the former, but not in the latter decision. This is noteworthy, considering that the gender effect was of similar strength in Study 1 (that concerned investment) and Study 2 (that concerned employment). In short, gender differences are robust to the factors that affect the daughter effect, but these are yet to be discovered. We need to point out that we are not the first to show no clear support for the daughter effect ([29]; however, see [30] for a methodological comment on that particular finding). Moreover, in one study, Dahl and colleagues [31]

showed that the birth of a child (even daughters, if the first-born child was not female) makes male CEOs less generous to employees.

The major implication from our research is that women are considerably less likely to consider employment in morally controversial companies than men. The reluctance of some people to perform immoral work causes compensation for immoral work to be higher [11]. The inhibitions of women to work for morally controversial companies would thus be a contributor to a gender gap in wages. Not that this is precisely what is happening for investments, as investment in 'sin stocks' is linked to higher expected returns [6]. It is reasonable to argue that an aversion towards working in morally controversial companies—at least in those people that have the opportunity to work in such companies—will have a much larger negative effect on a person's wealth than the lost excess returns that a person would enjoy from investment in such companies.

Opposition towards morally controversial stocks is economically meaningful, given that the proportion of stocks that might be classified as morally controversial is non-negligible, as are the wage premiums that they might offer to their employees. One estimate is provided by Trinks and Scholtens [32], who show that companies involved in the 14 morally controversial issues that they studied comprise 10% of the stocks of companies from the S&P 500 index, accounting for 12% of the market capitalization. Schneider and colleagues [11] use data from a representative sample of Swiss employees to estimate the premium enjoyed by those that do decide to work in such companies. They estimate that wages in tobacco companies are 35% higher than wages in neutral (conventional) companies. This, in our view, can serve as a plausible upper-bound for how much more socially controversial companies pay their employees to offset the immorality component of being employed there.

We should point out that the industry the company operates in is not the sole determinant of whether it is considered an immoral working place or target of investment. A company can also be deemed morally controversial for the practices that it perpetuates, creating an unethical working culture. To illustrate, in a poll of 14,500 employees in US companies, 23% of them stated that they sometimes or always experience the pressure to behave unethically in their workplace [33]. Facebook Inc. has been recently accused by a whistleblower of perpetuating a business model that hurts the mental health of its teenage users [34], which is yet another controversy that the company is involved in, that has probably extended the number of people reluctant to invest in or work for this company. Alternatively, people could base their employment decisions on the publicly available ratings of environmental, social and governance (ESG) issues. For example, they can decide that they will not seek employment in companies that have a low ESG rating (e.g., have a B or CCC rating in ESG ratings provided by MSCI) relative to its peers (i.e., companies from the same industry). At the time of writing, Facebook Inc. had a 'B' rating, making it one of the 29% of companies that were classified by MSCI as 'laggards' in the interactive media & services industry (note, however, that there is considerable variation in how the same companies are rated by different agencies [35]). Taken together, 10–30% of public or large companies could be plausibly classified as morally controversial. This can serve as a rough estimate of the proportion of companies that people might feel uncomfortable working in: if the findings of our study hold, the degree of discomfort should be higher in women.

The most important limitation to our study is that we tested judgments and not decisions. It is easier to judge something as immoral, than to refuse benefiting from an immoral course of action. Hence, e.g., even the most robust effect we report—the sin stock bias in women—could be attenuated had the stakes been real and high. Additionally, note that the daughter effect in investment was not present in all specifications, which creates some reservations concerning the robustness (or the strength) of the effect. If, however, the daughter effect

generalizes across decisions, but is small, it is possible that our employment study suffered from insufficient statistical power. Hence, we could have not observed the true daughter effect in the job-selection decisions merely due to chance.

Yet, we should point out that it is unlikely that the gender effect that we observed was due to a difference in hypothetical bias between men and women. A hypothetical bias is present if people are more likely to state a preference in a hypothetical scenario, which doesn't translate into actual behavior in the real-world (e.g., [36]). Evidence on the existence of gender differences in this bias are mixed [37, 38], but are more consistent with a higher hypothetical bias in men than in women, if these differences do exist. If so, it is plausible that real-life differences in investment behavior would be greater than the ones observed here.

A caveat of our analysis is that participants made simplified judgments about investment, having to rate the notion of making an investment (the extensive margin), and not the intensity of the investment (the intensive margin). It is plausible that differences in the framing of the investment question—e.g., by contrasting the lack of an investment, a negligible investment (0.01%) and a significant investment (10%), or a default investment (10%) with a doubled investment (20%)–could translate into stronger (or weaker) gender differences or daughter effects. Also, in our task, participants assessed each industry separately, which assumed that they excluded the other industries when making their assessment: in the real-world, people are more likely to decide between portfolios that pool companies from many industries (e.g., pool companies from the triumvirate of sin—alcohol, tobacco, and gambling [6]), and do not decide on each industry independently.

## 5. Conclusions

In this research, we replicated and extended the research on the factors affecting moral-economic decision making. Specifically, we suggest that women are reluctant to invest their capital—both their financial capital and their human capital—in morally controversial companies (sin stocks), even though the gains from the latter appear to be considerable. We also showed that parenting a daughter increases bias against sin stocks in men, but that this effect is fragile and far less robust than the corresponding gender difference. We hypothesize that the aversion towards morally controversial companies that women exhibit contributes to the gender gap in wealth and income, both in terms of the returns from portfolio investments, and in wages.

## Supporting information

**S1 Appendix. Experimental instructions.**
(PDF)

**S1 Table. Summary statistics.**
(PDF)

**S2 Table. The effect of the number of daughters.**
(PDF)

**S3 Table. Analysis of male and female subsamples: A comparison of the effect of having daughters, and the number of daughters conditional on the number of children.**
(PDF)

**S4 Table. Test of female socialization hypothesis and effect of first-born child.**
(PDF)

**S5 Table. Influence of differences in household income levels.**
(PDF)

**S6 Table. Judgment of individual industries (Studies 1 and 2 pooled).**
(PDF)

## Author Contributions

**Conceptualization:** Paweł Niszczota, Michał Białek.

**Data curation:** Paweł Niszczota.

**Formal analysis:** Paweł Niszczota.

**Funding acquisition:** Paweł Niszczota.

**Investigation:** Paweł Niszczota, Michał Białek.

**Methodology:** Paweł Niszczota, Michał Białek.

**Project administration:** Paweł Niszczota.

**Resources:** Paweł Niszczota.

**Software:** Paweł Niszczota.

**Supervision:** Paweł Niszczota.

**Validation:** Paweł Niszczota, Michał Białek.

**Visualization:** Paweł Niszczota.

**Writing – original draft:** Paweł Niszczota.

**Writing – review & editing:** Michał Białek.

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
