## [Decision Letter · Decision Letter 0]

3 Sep 2021

PONE-D-21-22149

The effect of gender and parenting daughters on judgments of morally controversial companies

PLOS ONE

Dear Dr. Niszczota,

Thank you for submitting your manuscript to PLOS ONE. I have now heard from one referee, several others whom I contacted where unable to provide a review. After my own reading of the paper and having heard from this one referee, who is giant in the field, I decided to invite you to revise and resubmit your paper. I agree with the issues raised by the referee and in my judgment, you should be able to constructively address all of them.

Please, make sure that your revision meets PLOS ONE’s publication criteria. Given that you run an experiment, please make sure that the content of your paper provides the reader with all the information necessary for them to judge the integrity of your research on their own.  

Please submit your revised manuscript by Oct 18 2021 11:59PM. If you need more time than this to complete your revisions, please reply to this message or contact the journal office at plosone@plos.org. Please include the following items when submitting your revised manuscript:

We look forward to receiving your revised manuscript.

Kind regards,

Joanna Tyrowicz

Academic Editor

PLOS ONE

Journal Requirements:

3. We note you have included a table to which you do not refer in the text of your manuscript. Please ensure that you refer to Table A.1, A.2, A.3, A.4, A.5 and A.6 in your text; if accepted, production will need this reference to link the reader to the Table.

Reviewers' comments:

Reviewer's Responses to Questions

**Comments to the Author**

1. Is the manuscript technically sound, and do the data support the conclusions?

Reviewer #1: Partly

2. Has the statistical analysis been performed appropriately and rigorously? 

Reviewer #1: Yes

3. Have the authors made all data underlying the findings in their manuscript fully available?

Reviewer #1: Yes

4. Is the manuscript presented in an intelligible fashion and written in standard English?

Reviewer #1: Yes

5. Review Comments to the Author

Reviewer #1: See the attached referee report

See the attached referee report

6. PLOS authors have the option to publish the peer review history of their article (what does this mean?). If published, this will include your full peer review and any attached files.

Reviewer #1: **Yes: **Marcin Kacperczyk

---

## [Author Response · Author response to Decision Letter 0]

19 Oct 2021

Specific Comments

1. My first problem reading the paper is that it omits some important information that would be useful to report when presenting the results. For example, the results from the regressions do not show the list of controls that are included and more so the coefficients of these variables. These would be useful as they would give a reader a better sense of the properties of the sample. Similarly, I did not find good evidence on the definition of controversial sectors. Perhaps this has been reported in the past study, but I think it would be necessary to bring up some of these materials to the current draft as well.

RESPONSE

We’d like to thank the Reviewer for these suggestions. Our tables were indeed unorthodox, in that they did not list which control variables were used, and what were their estimates. We have amended this: now all our tables (including the supplementary tables) include estimates for all control variables.

We now also mention in the text which controversial and conventional stocks we used, additionally pointing the reader to descriptions (in the Appendix). This should have been indicated in the earlier version of the paper – we appreciate that the Reviewer has pointed out this issue, as it enhances the readability of the manuscript.

2. Like the authors, I am also worried a bit about the Rabin effect, that is, whether intent to do something is equivalent to the act of doing it itself. Is there some evidence that the authors could bring up that this distinction is gender or gender/offspring indifferent? I think this would be a partial way to address the concern whether one can interpret the results as economically meaningful.

RESPONSE

This is an excellent observation. Indeed, the differences that we observed could potentially be the result of gender differences in hypothetical bias, i.e. the difference between the intent to act and actual action. However, there is no consistent evidence on whether this bias is different between men and women (Johansson-Stenman & Svedsäter, 2012): significant differences, if present, mostly show that the hypothetical bias is actually lower in women. This would suggest that women – relative to men – state preferences that are not that far away from real-life (revealed) preferences (in our case: a preference of not investing in morally controversial companies, and not wanting to be employed by them). (We are unaware of any studies that would investigate how the gender of children might impact hypothetical bias.)

We address this issue in the Discussion.

3. It would be helpful to clarify that the results from both experiments are extensive margin results, which do not allow for the possibility of the variation in intensity of action. This is particularly true for investment decisions. In some sense, the experiments assume the strict exclusionary screening on its participants.

RESPONSE

Another fine point. Indeed, we designed both studies with the intention of studying extensive margins. We now confirm that this is the case in the Introduction. We also point out that we assumed that participants would be able to treat our task as one that relates to extensive margins, and not intensive margins (the latter could be especially relevant if they actually invested or were employed in companies from the listed industries). Participants were assumed to be robust to such cases, indeed being able to correctly infer that we were after the extensive margin.

We now notice in the Discussion that this is an important distinction, and that studying intensive margins results could potentially lead to different results.

4. It would be useful to provide some discussion of the economic magnitudes. Can you link your results to some form of aggregates? When you discuss issues like gender gap, how should we think of these ideas in the context of the fraction of jobs that are available in the controversial industries, average pay in these jobs, etc?

RESPONSE

This is a very useful comment. It is important to point out to the reader that this is actually a salient issue, given that the proportion of companies that could be considered as morally controversial is considerable. We provide a (reasonable, in our view) lower and upper bound for the part of companies that could be considered ‘sinful’ in some way. Firstly, we point to data from Trinks and Scholtens (2017), who estimate that companies dealing with the fourteen controversial issues they study constitute 10% of the number of S&P 500 stocks, and 12% of the market capitalization of companies from that index. But this is just a lower bound, given the growing number of issues that could contribute to the company being treated as ‘sinful’. Secondly, we point the reader to a US survey consisting of over 14,500 employees, in which 23% of them indicated that (Ivcevic et al., 2020). Lastly, we suggest that ESG ratings could be used as proxies for being sinful (e.g., if a company is classified as a ‘laggard’ by MSCI).

We also provide estimates for the premium resulting from working for morally controversial companies, that are provided Schneider et al. (2020). In that study, it was estimated that tobacco companies pay a 35% higher wage than neutral (conventional) companies. Given that tobacco companies are (probably) one of the most controversial ones, this (probably) serves as an upper bound for the effect (which we acknowledge in the text). This should be a useful proxy for the reader, nonetheless.

5. In the discussion section, the authors argue that their results suggest that men might internalize the preferences of their daughters. I find this result quite surprising. Does this mean the daughters are already grown-up individuals? If so, it would be useful to bring up some evidence on the demographics of these daughters/sons. My interpretation would be that having a daughter may carry a different emotional load and men may want to cater to this social norm pressure. Can the authors rule out such explanation? What is the basis for such inference?

RESPONSE

This is a great observation, something that indeed requires clearing up. We assume that the Reviewer is referring to the following, indeed misleading sentence from our discussion: “This effect suggests that men might internalize the preferences of their daughters, making their judgments and behaviors more in line with those manifested by women”. In truth, we actually believe in the interpretation provided by the Reviewer, that being a parent to a daughter causes parents to behave differently (due to, e.g., expected or actual social pressures). It should not require parents to observe the behavior of their own child, and instead be the result of a quick adaptation to the reality of being a parent to a daughter. Pogrebna et al. (2018) provide some evidence that this is the case, using data from future parents who have just found out about the sex of their not-yet-born child. Our own results (reported in Table S4) are also generally consistent with this interpretation, suggesting that for the daughter effect to work, it does not require years of socialization (parents “growing up” with their children).

We have now amended the Introduction, to make clear that this is our (and the Reviewer’s) assumption concerning the mechanism behind potential daughter effects.

---

Overall, we are indebted to the Reviewer for finding the time to review our manuscript. This is kind. The comments are insightful, and have led to revisions that the readers will benefit from.

We’d also like to take the opportunity to thank the Reviewer for the pioneering work that has inspired not only this paper, but a whole research project that we are involved in. Thank you!

References

Ivcevic, Z., Menges, J. I., & Miller, A. (2020, March 20). How Common Is Unethical Behavior in U.S. Organizations? Harvard Business Review. https://hbr.org/2020/03/how-common-is-unethical-behavior-in-u-s-organizations

Johansson-Stenman, O., & Svedsäter, H. (2012). Self-image and valuation of moral goods: Stated versus actual willingness to pay. Journal of Economic Behavior & Organization, 84(3), 879–891. https://doi.org/10.1016/j.jebo.2012.10.006

Pogrebna, G., Oswald, A. J., & Haig, D. (2018). Female babies and risk-aversion: Causal evidence from hospital wards. Journal of Health Economics, 58, 10–17. https://doi.org/10.1016/j.jhealeco.2017.12.006

Schneider, F., Brun, F., & Weber, R. A. (2020). Sorting and Wage Premiums in Immoral Work (SSRN Scholarly Paper ID 3646150). Social Science Research Network. https://doi.org/10.2139/ssrn.3646150

Trinks, P. J., & Scholtens, B. (2017). The Opportunity Cost of Negative Screening in Socially Responsible Investing. Journal of Business Ethics, 140(2), 193–208. https://doi.org/10.1007/s10551-015-2684-3

---

## [Decision Letter · Decision Letter 1]

11 Nov 2021

The effect of gender and parenting daughters on judgments of morally controversial companies

PONE-D-21-22149R1

Dear Dr. Niszczota,

We’re pleased to inform you that your manuscript has been judged scientifically suitable for publication and will be formally accepted for publication once it meets all outstanding technical requirements.

Kind regards,

Joanna Tyrowicz

Academic Editor

PLOS ONE

Additional Editor Comments (optional):

Reviewers' comments:

Reviewer's Responses to Questions

**Comments to the Author**

1. If the authors have adequately addressed your comments raised in a previous round of review and you feel that this manuscript is now acceptable for publication, you may indicate that here to bypass the “Comments to the Author” section, enter your conflict of interest statement in the “Confidential to Editor” section, and submit your "Accept" recommendation.

Reviewer #1: All comments have been addressed

2. Is the manuscript technically sound, and do the data support the conclusions?

Reviewer #1: Yes

3. Has the statistical analysis been performed appropriately and rigorously? 

Reviewer #1: Yes

4. Have the authors made all data underlying the findings in their manuscript fully available?

Reviewer #1: Yes

5. Is the manuscript presented in an intelligible fashion and written in standard English?

Reviewer #1: Yes

6. Review Comments to the Author

Reviewer #1: The authors have successfully addressed all my questions and I have no further comments on the manuscript. I suggest that the paper be accepted for publication.

7. PLOS authors have the option to publish the peer review history of their article (what does this mean?). If published, this will include your full peer review and any attached files.

Reviewer #1: **Yes: **Marcin Kacperczyk

---

## [Editor Report · Acceptance letter]

18 Nov 2021

PONE-D-21-22149R1 

The effect of gender and parenting daughters on judgments of morally controversial companies 

Dear Dr. Niszczota:

I'm pleased to inform you that your manuscript has been deemed suitable for publication in PLOS ONE. Congratulations! Your manuscript is now with our production department. 

Kind regards, 

on behalf of

Professor Joanna Tyrowicz 

Academic Editor

PLOS ONE